# Protocol Optimization for Direct Reprogramming of Primary Human Fibroblast into Induced Striatal Neurons

**DOI:** 10.3390/ijms24076799

**Published:** 2023-04-05

**Authors:** Nina Kraskovskaya, Anastasia Bolshakova, Mikhail Khotin, Ilya Bezprozvanny, Natalia Mikhailova

**Affiliations:** 1Center of Cellular Technologies, Institute of Cytology of the Russian Academy of Science, 194064 St. Petersburg, Russia; ninakraskovskaya@gmail.com (N.K.);; 2Laboratory of Molecular Neurodegeneration, Peter the Great St. Petersburg Polytechnic University, 195251 St. Petersburg, Russia; 3Department of Physiology, UT Southwestern Medical Center at Dallas, Dallas, TX 75390, USA

**Keywords:** direct reprogramming, cell synchronization, medium spiny neurons, lentiviruses, microRNA, aging, dendritic spines

## Abstract

The modeling of neuropathology on induced neurons obtained by cell reprogramming technologies can fill a gap between clinical trials and studies on model organisms for the development of treatment strategies for neurodegenerative diseases. Patient-specific models based on patients’ cells play an important role in such studies. There are two ways to obtain induced neuronal cells. One is based on induced pluripotent stem cells. The other is based on direct reprogramming, which allows us to obtain mature neuronal cells from adult somatic cells, such as dermal fibroblasts. Moreover, the latter method makes it possible to better preserve the age-related aspects of neuropathology, which is valuable for diseases that occur with age. However, direct methods of reprogramming have a significant drawback associated with low cell viability during procedures. Furthermore, the number of reprogrammable neurons available for morphological and functional studies is limited by the initial number of somatic cells. In this article, we propose modifications of a previously developed direct reprogramming method, based on the combination of microRNA and transcription factors, which allowed us to obtain a population of functionally active induced striatal neurons (iSNs) with a high efficiency. We also overcame the problem of the presence of multinucleated neurons associated with the cellular division of starting fibroblasts. Synchronization cells in the G1 phase increased the homogeneity of the fibroblast population, increased the survival rate of induced neurons, and eliminated the presence of multinucleated cells at the end of the reprogramming procedure. We have demonstrated that iSNs are functionally active and able to form synaptic connections in co-cultures with mouse cortical neurons. The proposed modifications can also be used to obtain a population of other induced neuronal types, such as motor and dopaminergic ones, by selecting transcription factors that determine differentiation into a region-specific neuron.

## 1. Introduction

The production of human neuronal cells in laboratory conditions became possible due to the discovery of induced pluripotency, which makes it possible to reprogram differentiated somatic cells into pluripotent ones (iPSc) [1]. The expression of the transcription factors Oct4 (Pou5f1), Sox2, Klf4, and c-Myc, which are responsible for maintaining pluripotency in embryonic stem cells, is capable of inducing pluripotency in somatic cells [1]. Obtained iPSC cells retain the ability to proliferate an unlimited number of times and the potential for differentiation in different cell types, including neuronal cells. Special differentiation media in combination with small molecules and neurotrophic factors allow us to obtain almost an unlimited number of particular types of human neuronal cells. Several iPSCs-based protocols have been published that model HD neuropathology in cultures [2,3,4,5]. Despite the great advantage of iPSC-derived neurons, they also have a disadvantage associated with the “youth” of the obtained cells. Neurodegenerative diseases, including HD, primarily are age-related pathologies. Upon transition to the pluripotent state, cells lose all the epigenetic marks that were acquired during the differentiation and maturation of the somatic cells from which they originated [6,7]. To become more mature, they need extended cultivation time or additional factors that contribute to cell aging, such as the shortening of telomere length [8] or moderate progirin expression [9]. Several recent studies have also proposed using isogenic cell lines with additional CRISPR/Cas9 CAG repeat correction to obtain a polyglutamine-dependent HD phenotype [10,11,12].

The problem of cell aging is partly solved by using a different approach based on direct reprogramming [13,14]. The main advantage of direct reprogramming methods is the preservation of the epigenetic information in cells confirmed by the “epigenetic clock” method, based on the assessment of the DNA methylation status [15]. Thus, due to the preservation of the age-associated phenotype, it is possible to study the pathophysiological features of the development of the disease at each stage of neuropathology.

Several different approaches have been proposed to stimulate the direct conversion of fibroblasts into induced neurons, such as the overexpression of transcription factors ASCL1, Brn2, and MYT1L [13] or the knockdown of RNA-binding protein PTB [16]. One of the most promising methods in terms of the effectiveness of the procedure itself turned out to be a combination of miRNAs and transcription factors that was proposed by Andrew Yoo and colleagues [14]. In this protocol, microRNAs are the main factors that stimulate chromatin remodeling to activate proneuronal genes. The addition of transcription factors MYT1L and ASCL1 enhances the process of neuronal differentiation. Further, the protocol was supplemented with DLX1/2 and CTIP2 transcription factors in order to generate DARPP32-positive induced medium spiny neurons, which are the most prone to the cell death in HD [17]. Then this protocol was applied to the study of HD pathogenesis. The authors showed that the technique of direct reprogramming preserves the age-associated disease phenotype, and the efficiency of reprogramming does not differ for fibroblasts between healthy donors and HD patients [18]. The authors compared the vulnerability to cellular stress in neurons reprogrammed from fibroblasts of healthy donors and HD patients at the pre-symptomatic and symptomatic stages. According to the obtained results, the neurons at the pre-symptomatic stage have a lower level of cell death and oxidative DNA damage than the cells at the symptomatic stage. Moreover, the main histopathological component of HD-mutant Huntingtin aggregates can be detected in reprogrammed striatal cells at both stages. Thus, the developed approach to the modeling of HD is extremely valuable and can serve as an important intermediate step between research on model organisms and clinical trials.

However, while setting up the protocol, we have noticed that after the reprogramming procedure, some of the cells contain more than one nucleus, which can lead to a change in the functional properties of the obtained neurons. The appearance of such cells is easily explained by the fact that fibroblasts are dividing cells and at the beginning of chromatin remodeling, after microRNA induction, they exist at different stages of the fibroblasts’ cell cycle. In order to synchronize the cells in the G1 phase after transduction with lentiviruses, we introduced an additional synchronization step by applying a rapamycin, a well-characterized inhibitor of G1 cell cycle progression [19].

We also decided to minimize the number of transcription factors used in the protocol because we observed that an extensive number of viruses is toxic and decreases cell survival during the reprogramming procedure. The authors used both DLX1 and DLX2 transcription factors to differentiate cells in GABAergic neurons. However, it was reported that DLX1 is expressed prenatally, but not neonatally, while adult parvalbumin-positive interneurons maintain DLX2 expression and DLX2 directly modulates the expression of essential GABAergic markers, such as GAD1/2 and VGAT [20]. Moreover, the expression of DLX2 along with ASCL1 induced GABAergic neuron differentiation from hiPSC-derived NPCs and from ES [21,22]. Based on these data, we assume that the overexpression of only DLX2 is sufficient to convert fibroblasts into GABAergic neurons.

As a result, we propose a modified protocol to ensure a high reprogramming efficiency and the homogeneity of iSNs in terms of their morphological and functional characteristics, capable of forming synaptic connections.

## 2. Results

### 2.1. Experimental Design and Efficiency of Modified Protocol

Despite all the advantages of the protocol developed by the group of Andrew Yoo, we encountered several issues in its application. The main issue was that, at the end of the reprogramming procedure, some cells had two or sometimes even more nuclei (Figure 1A). In order to overcome this issue, we added a synchronization step before the induction of microRNA expression with doxycycline (Figure 1B). As a result of the synchronization, we observed a decrease in the number of actively dividing cells, which was confirmed by a decrease in the number of cells in the S phase from 12.17 ± 0.4% to 3.75 ± 0.18% (** *p* < 0.01) (Figure 1C). As a result, the number of multinuclear cells decreased to 2.9 ± 1.42% for the modified protocol, while for the original protocol, the number of multinucleated cells was 10.32 ± 1.61% (** *p* < 0.01) (Figure 1D).

Since we did not make principal changes to the original protocol, we observed similar morphological changes in the cells during the reprogramming procedure (Figure 2A). Cell cycle synchronization and the reduction in virus number resulted in an increase in the survival of neurons at each stage of selection. After the first and second step of antibiotic selection, the fibroblasts were fixed and stained with DAPI (Figure 2B). The number of surviving cells was counted in the Image_J software. The number of synchronized untransduced cells was taken as 100%. After one selection, the survival rate was 92.3 ± 5.9%, and after two selections, the survival rate was 85.5 ± 8.1%. By the end of the reprogramming procedure, the number of surviving cells did not significantly decrease compared with the number of cells after the second selection (Figure 2C). By the end of the reprograming procedure (typically 5 weeks after the transduction with lentiviruses), the vast majority of the cells were positively stained for neuronal markers TUJ1 and MAP2, with 94.7 ± 3.7% and 94.1 ± 6.7% stained cells, respectively (Figure 2D,E). Even though DARPP-32 staining on 36 PID was detected in 90.1 ± 9.3% of the cells, it was mostly restricted to soma.

### 2.2. Functional Properties and Morphology of Induced Neurons

In addition to immunofluorescence staining, the specificity of reprogramming cells into neuronal cells was confirmed by Ca^2+^ imaging experiments. Reprogrammed neurons were loaded with the chemical fluorescent calcium indicator Fluo-4 (Figure 3A). The application of 56 mM of a potassium chloride solution, which causes membrane depolarization, is characteristic only of excitable cells, such as neurons. As a result of the experiment, 88.89 ± 9.07% responded to stimulation. In order to confirm the ability of cells to respond to excitory signals, the cells were stimulated with 100 μM of a glutamate solution, and 99.33 ± 4.62% of cells responded to the application of the neurotransmitter (Figure 3B,C).

The induced neurons obtained by direct reprogramming are characterized by a complex morphological structure inherent to striatal neurons and have a branching dendritic tree (Figure 3D). It is also important to note the absence of dendritic spines in neurons, which additionally characterizes the homogeneity of the resulting population. In monocultures, GABAergic neurons do not form dendritic spines, since most inhibitory synapses are formed on dendritic shafts and cell bodies [23].

### 2.3. Formation of Dendritic Spines on iSNs in Co-Culture with Mouse Cortical Neurons

The striatum receives glutamatergic signals from the entire cortex and nuclei of the thalamus, as well as dopaminergic signals from dopamine neurons of the substantia nigra, representing the main integrator of information in the basal ganglia system [24]. As a result, a large number of dendritic spines—postsynaptic structures of excitatory synapses—are formed on the dendritic tree of striatal neurons [25]. The evaluation of the number and morphology of dendritic spines is important for analyzing functional changes in neurons that occur during the development of HD [11,26,27,28,29,30,31,32]. The ability of a particular drug to maintain synaptic activity may indicate its neuroprotective effect [26,28].

However, in monocultures, striatal neurons do not form dendritic spines, which are intrinsic to this type of cells under physiological conditions [33]. In order to confirm the ability of iSNs to form synaptic connections, we inoculated mouse cortical neurons isolated from wild-type pups to the iSNs to resemble a cortico-striatal co-culture. In order to distinguish mouse cortical neurons from iSNs, after the end of the procedure, the induced neurons were transduced with a lentivirus encoding the red fluorescent protein mCherry. Figure 4A shows an iSN in a co-culture with cortical neurons isolated from newborn mice and stained for the neuronal synaptic marker synapsin1 on the 14^th^ day of co-cultivation. As expected, in the absence of cortical neurons, iSNs do not form dendritic spines (0 DIV). Co-cultivation with cortical neurons stimulates filopodia formation, which appears as early as at 7 DIV. And by 10 DIV the dendritic spines start to form on the dendrites of iSNs, which indicates the formation of synapses between the two types of neurons (Figure 4B).

## 3. Discussion

Nowadays, the vast majority of studies on the pathogenesis of HD are carried out on the induced striatal cells of patients obtained with iPSC technology [34,35,36]. This is largely due to the existence of reliable protocols that make it possible to obtain a large population of neuronal cells. iPSCs have an almost unlimited potential for self-renewal. Therefore, having received a several clones of iPSc, it is possible to expand them and, after the induction of neuronal differentiation, obtain a large population of DARPP-32-positive cells. Hence, the low conversion efficiency into iPSCs is not a limiting factor for working with a large pool of the neurons. However, in the study of the age-related mechanisms of HD and other age-related pathologies, such as Alzheimer’s disease and Parkinson’s disease, direct reprogramming methods are extremely useful since they allow us to bypass the pluripotent stage and thereby preserve age-related changes in induced neurons [15]. However, the cell cycle exit precedes the differentiation into neurons, which makes it impossible to increase the populations of induced neuronal cells at the end of the procedure. Therefore, the problem of the effectiveness of the procedure when working with direct methods of reprogramming imposes a significant restriction on its application in research studies. This is due to the fact that the number of neuronal cells available for morphological and functional studies after the reprogramming procedure are limited by the initial number of fibroblasts. Despite the prospects for the preservation of epigenetic information embedded in cells, the main disadvantage of direct reprogramming methods is their low efficiency and significant cell loss during the procedure.

Trying to answer the question whether there are any fundamental features that limit the transdifferentiation of fibroblasts into neuronal cells, the authors assumed that the main contribution was the heterogeneity of initial cells due to the cell cycle [37]. Neurons are believed to have permanently blocked their capacity to proliferate once they are differentiated, being typically found in a quiescent state in the adult nervous system. Meanwhile, dermal fibroblasts obtained from young and middle-aged donors, an age when HD manifests, represent a population of actively dividing cells [38]. Moreover, the results from another study group have demonstrated that the attenuation of p53, in conjunction with cell cycle arrest at G1 obtained by serum starvation and an appropriate cell culture environment, increases the efficiency of the transdifferentiation of human fibroblasts to induced dopaminergic neurons [39]. Taking into consideration our observations on the presence of multinucleated neurons, we assumed that the synchronization of cells in the G1 phase will not only decrease the amount of multinucleated cells but also will positively affect the efficiency of the reprogramming procedure.

In order to synchronize cells, we used rapamycin-induced cell cycle arrest. We found that fibroblast synchronization promoted reprogramming efficiency up to 80% from the starting cells and prevented the appearance of multinucleated neurons due to the vast majority of cells that were restricted from cell cycle progression. We used rapamycin because serum starvation is often insufficient for effective cell synchronization, and the addition of another lentivirus to knockdown p53 would reduce overall cell survival.

We also demonstrated that the combination of the transcription factors DLX2, CTIP2, and MYT1L is sufficient to obtain a homogeneous population of iSNs positive for DARPP32, and the resulting cells retain a complex morphological structure and respond to stimulation with potassium chloride and glutamate. Due to the increase in homogeneity, the use of a modified protocol allows us to reduce variability in morpho-functional studies.

Finally, we showed that reprogrammed neurons are capable of forming synaptic connections. The idea of the co-cultivation of striatal and cortical neurons for the formation of dendritic spines in striatal neurons was first proposed and studied in detail by M. Segal and colleagues [33]. The technique of the co-cultivation of the neurons of the cortex and striatum most fully reflects the formation of synaptic connections in the brain, since the striatum in vivo receives afferent innervation mainly from cortical neurons. The inoculation of cortical neurons to the striatal neurons’ monoculture resulted in both spontaneous and induced neuronal activity, as well as a 10-fold increase in the density of dendritic spines. The presence or absence of cortical cells does not affect the survival rate of MSN. During co-cultivation, induced striatal neurons form synaptic contacts with cortical neurons, which allows cortical neurons to carry out synaptic transmission to striatal neurons, which, in turn, respond with electrophysiological activity and form numerous synaptic contacts. In our study, iSNs were capable of forming synaptic connections with mouse cortical neurons and forming dendritic spines. The assessment of their number and morphology is an important tool for analyzing functional changes in neurons that occur during HD progression, as evidenced by the results of several studies [26,28,30]. The proposed protocol modifications can also be used to obtain homogeneous populations of other types of neurons, such as motor and dopaminergic ones, by selecting specific transcription factors that determine differentiation into a particular cell type.

## 4. Materials and Methods

### 4.1. Cell Lines

Primary human dermal fibroblasts from 2 women who were 37 and 42 years old were obtained from Collection of Vertebrate Cell Cultures Institute of Cytology RAS, Saint Petersburg. Primary human dermal fibroblasts from a 41-year-old man were obtained from Collection of Cell Cultures for Biotechnological and Biomedical Research Institute of Developmental Biology RAS, Moscow.

### 4.2. Plasmids and Reagents

rtTA-N144 (hygromycin mammalian resistance; Addgene, 66810)

pTight-9-124-BclxL (puromycin mammalian resistance; Addgene, 60857)

pmCTIP2-N106 (blasticidin mammalian resistance; Addgene, 66808)

phDLX2-N174 (neomycin mammalian resistance; Addgene, 60860)

phMYT1L-N174 (neomycin mammalian resistance; Addgene, 66809)

Special Reagents:-Dibutyryl-cAMP sodium salt (Sigma-Aldrich, St. Louis, MI, USA, Cat. No. D0627)-Valproic acid sodium salt (VPA) (Sigma-Aldrich, St. Louis, MI, USA, Cat. No. P4543)-Recombinant human brain-derived neurotrophic factor (BDNF) (Peprotech, Cranbury, NJ, USA, Cat. No. 450-02)-Recombinant human neurotrophin-3 (NT3) (Peprotech, Cranbury, NJ, USA, Cat. No. 450-03)-Retinoic acid (RA) (Sigma-Aldrich, St. Louis, MI, USA, Cat. No. R2625)-Matrigel (Corning Incorporated, Corning, NY, USA, Cat. No. 354277),-Polybrene 10 mg/mL stock (Merck Millipore, Burlington, MA, USA, Cat. No. TR-1003-G)-Rapamycin (Selleckchem, Planegg, Germany, Cat. No. AY-22989)

Antibiotics:-Doxycycline (DOX) (Sigma-Aldrich, St. Louis, MI, USA, Cat. No. D3072)-Puromycin (Thermoscientific, Waltham, MA, USA, Cat. No. A1113803)-Blasticidin (Thermoscientific, Cat. No. A1113903)-G418 (Sigma-Aldrich, St. Louis, MI, USA, Cat. No. G8168)

Medium:-Neurobasal-A (Gibco, Thermoscientific, Waltham, MA, USA, Cat. No. 10888022)-DMEM (Gibco, Thermoscientific, Waltham, MA, USA, Cat. No. 11965092)-FBS (Gibco, Thermoscientific, Waltham, MA, USA, Cat. No. A3160802)-GlutaMAX supplement (Gibco, Thermoscientific, Waltham, MA, USA, Cat. No., 35050061)-0.25% trypsin-EDTA (Gibco, Thermoscientific, Waltham, MA, USA, Cat. No. 25200056)-DPBS (Gibco, Thermoscientific, Waltham, MA, USA, Cat. No. 14190144)

HEK293T medium: DMEM +10% FBS

Fibroblast medium (FM): DMEM +10% FBS + 1xGlutamax

Reprogramming media (RM): Neurobasal-A + 1xB-27+ 0.5 mM glutaMAX + 1 mM VPA + 200 µM db cAMP + 20 ng/mL human BDNF +20 ng/mL human NT-3 + 1 µM RA. Protect from light, filter 0.22 before use.

### 4.3. Lentivirus Production

Prepare 100 mm Petri dish of HEK293T line cells with 70–80% confluency. The production of lentiviruses was carried out in HEK 293T cells by co-transfection of the pPAX packaging vector, the pMD2.G delivery vector, and the shuttle plasmid in a 3:1 ratio with four using polyethyleneimine at a concentration of 1 mg/mL. After 16 h, the cells were washed with PBS and the medium was changed to fresh HEK 293T cell medium. After 48 h of co-transfection, the conditioned medium in which HEK293T cells were incubated was collected and centrifuged at 2500 rpm for 5 min. After that, the conditioned medium containing viral particles was passed through a PES filter with a pore diameter of 0.45 µm. Concentrate lentiviral particles by ultracentrifugation at 70,000× *g* at a temperature of +4 °C for 2 h. Titration of lentiviruses was performed on primary dermal fibroblasts selected with appropriate antibiotics 48 h after transduction. Detailed information about reprogramming can be found in the Appendix A section.

### 4.4. Reprogramming Procedure

The procedure for reprogramming fibroblasts in iSN was performed according to the protocol with subsequent modification described in detail in the Appendix A section.

### 4.5. Flow Cytometry

Primary dermal fibroblasts (1 × 10^6^) with or without cell cycle synchronization were harvested by trypsinization and washed twice with (PBS). The cells were fixed in 10% formalin. The number of cells in S phase was detected with EdU proliferation kit (Abcam, Cat. No. ab222421). The cells’ staining was detected by flow cytometry (Cytoflex, Beckman Coulter, Brea, CA, USA), and the results were analyzed with Cytexpert 2.4 software to assess the percentage of cells in the S phases.

### 4.6. Immunofluorescence

Before immunofluorescence staining, the cells were washed from the conditioned medium with phosphate-buffered saline (PBS) and fixed in a solution containing 10% formaline (Sigma-Aldrich, Cat. No. HT5011) for 20 min at room temperature. Then a series of PBS washes were performed and the cell membranes were permeabilized with a 0.25% Triton X-100 solution in PBS for 5 min at room temperature. Then the cells were washed with PBS 3 times and incubated in a blocking solution containing 5% bovine serum albumin (Sigma-Aldrich, Cat. No. A9403) diluted in PBS for 1 h at room temperature to prevent nonspecific antibody binding. Then the cells were incubated with primary antibodies diluted in a blocking solution at a concentration indicated in Table 1 overnight at a temperature of +4 °C. Then a series of washes were performed with a PBS solution, after which the cells were incubated with secondary antibodies for 1 h at room temperature, diluted in blocking solution to the concentration indicated in Table 1. Then a series of washes were performed with a PBS solution and stained with DAPI, after which the cells were fixed on a slide using Mounting Medium with DAPI (Cell Signalling, Danvers, MA, USA, Cat. No. 8961S). Images were captured with Olympus Fluoview FV3000 laser confocal system.

### 4.7. Ca^2+^ Imaging

Fluo-4 Ca^2+^- imaging experiments with reprogrammed iSNs were performed on 35 PID. Cells were loaded with 5 µM Fluo-4 (Invitrogen, Waltham, MA, USA, Cat. No. F14201) for 30 min at 37 °C in condition media in the presence of 0.05% Pluronic F-127 (Sigma Aldrich, Cat. No. P2443). Reprogrammed iSN were stimulated with 56 mM KCl solution and 100 µM glutamate solution (Sigma Aldrich, Cat. No. G5889). The Fluo-4 fluorescent signal was recorded using an Olympus IX70 inverted epifluorescent microscope equipped with a 40× water lens (Olympus, Tokyo, Japan), a Zyla 4.2 digital camera (Andor Technology Ltd, Oxforf, UK), and a ThorLabs LED. Fluo-4 was excited by a LED at a wavelength of 470 nm. The experiments were controlled by the Micro-Manager 2.0 imaging software package. The resulting series of images were saved in TIFF format and further processed in the Image_J software. The numerical values of the fluorescence intensity in relative units in each time interval equal to 1 s were exported to the Excel program from the MS Office package. Values are presented as F/F_0_ time curves, where F is the measured fluorescence intensity of Fluo-4, and F_0_ is the baseline fluorescence intensity.

### 4.8. Primary Mouse Neuronal Cultures

Primary cortical neuronal cultures were prepared from wild-type pups of FVBN/NJ strain (Jackson laboratory, Bar Harbor, ME, USA, Cat. No. 001800). Briefly, regions of the cortex of postnatal day 0 mouse pups were dissected in ice-cold Hanks’ Balanced Salt Solution, digested with papain solution (30 min at 37 °C; Worthington, Columbus, OH, USA Cat. No. 3176) and dissociated with 1 μg/mL DNase I (Sigma-Aldrich, St. Louis, MI, USA, Cat. No. DN-25) solution. Cortical neurons were cultivated in medium containing Neurobasal-A, supplemented with 2% B-27, 5% FBS, and 0.5 mM GlutaMAX. At DIV7, half of the medium without FBS was added to each well.

### 4.9. Statistical Analysis

The results are presented as the mean ± standard error of the mean. Statistical comparison of experimental results was performed using Student’s *t*-test to compare two groups. The *p* values are indicated in the figure captions. The results were considered significant if the probability (*p*) of violating the hypothesis of no differences between the samples was less than 0.05.

## Figures and Tables

**Figure 1 ijms-24-06799-f001:**
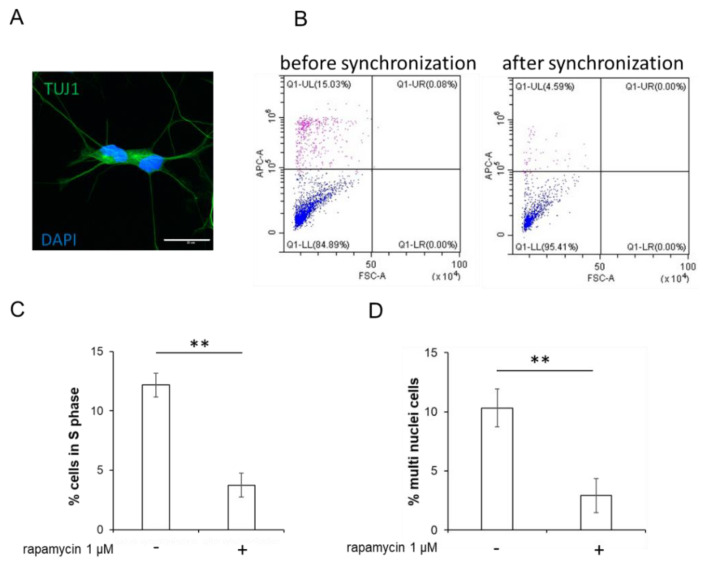
Cell cycle synchronization and experimental design of modified protocol. (**A**) Representative image of the neuron, obtained with original protocol and containing two nuclei. Immunofluorescent staining of induced neuron with antibodies for the neuronal marker protein TUJ1 (secondary antibodies conjugated with the Alexa 488 fluorophore). DAPI was used for identification of cell nuclei. Confocal microscopy, ×60. The scale bar corresponds to 20 μm. (**B**) Detection of divided cells by flow cytometry analysis of DNA content in primary human dermal fibroblasts stained with Edu kit and not subjected to (−) or subjected (+) to the cell cycle synchronization with 1 µM rapamycin for 48 h. (**C**) Percentages of cells in the S phases of the cell cycle without or with synchronization for 48 h with rapamycin. Data are presented as the mean ± SEM. ** *p* < 0.01. Student’s *t*-test. (**D**) Percentages of multinucleated neurons at the end reprogramming procedure reprogrammed according to the original and modified protocol. *n* = 272 neurons for original protocol, *n* = 330 neurons for modified protocol. ** *p* < 0.01. Student’s *t*-test.

**Figure 2 ijms-24-06799-f002:**
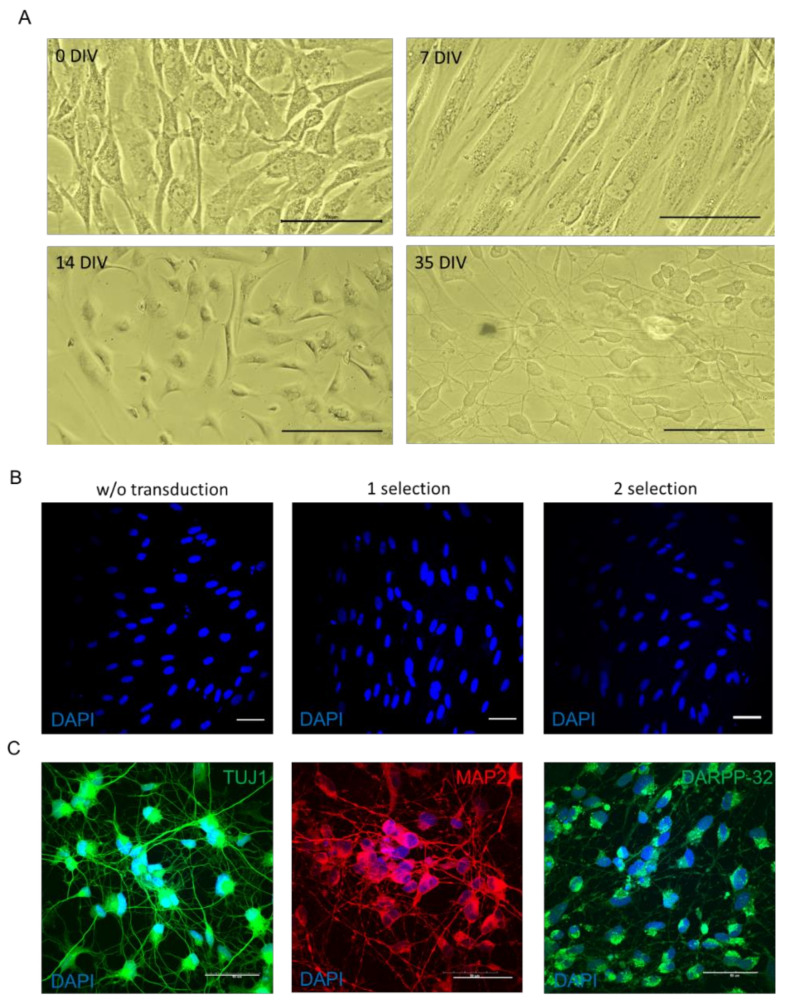
The effectiveness of modified reprogramming protocols. (**A**) Images illustrating morphology of induced neuron at 0 DIV, 7 DIV, 14 DIV, and 35 DIV. Light microscopy. Scale bars correspond to 100 µm. (**B**) Representative images of fibroblasts stained with DAPI without transduction of lentiviruses (*w*/*o*), after 1 selection with antibiotics, and after re-plating on coverslips and secondary selection with antibiotics, according to the modified protocol. Scale bar corresponds to 50 μm. (**C**) Representative images of iSNs reprogrammed according to the modified protocol and stained with antibodies to the neuronal markers and counterstained with DAPI. Immunofluorescent staining of induced neurons with antibodies for the neuronal marker protein TUJ1 (secondary antibodies conjugated with the Alexa 488 fluorophore), MAP2 (secondary antibodies conjugated with the Alexa 594 fluorophore), and DARPP-32 (secondary antibodies conjugated with the Alexa 488 fluorophore). Scale bar corresponds to 50 μm. (**D**) Histogram represents the number of remaining cells without transduction of lentiviruses (w/o) after the first and second steps of selection. Data are presented as the mean ± SEM. *n* > 500 cells. (**E**) Histogram represents the number of the TUJ1-, MAP2-, DARPP-32-positive cells counterstained with DAPI and reprogrammed according to the modified protocol. Data are presented as the mean ± SEM. *n* > 250 cells for each staining.

**Figure 3 ijms-24-06799-f003:**
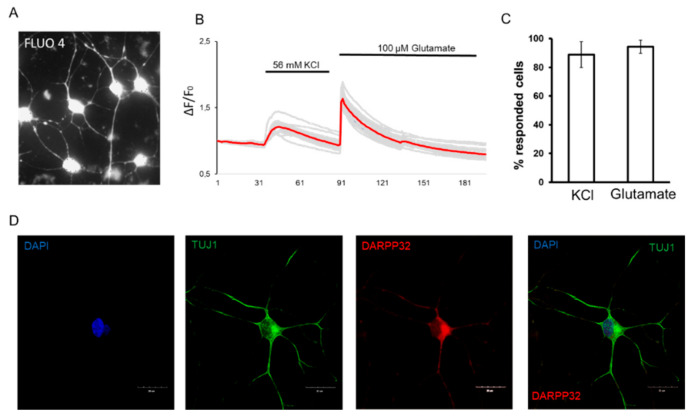
Functional assay study for reprogrammed iSNs. (**A**) Representative image of reprogrammed iSNs loaded with Fluo-4. (**B**) Cytosolic calcium cation levels in reprogrammed iSNs in response to depolarization of the plasma membrane with 56 mM potassium chloride (KCl) and 100 µM glutamate solution. Black lines on the top of the graph indicate the moment of application of 56 mM KCl and 100 µM glutamate solution to the cells. The gray curves illustrate the response of an individual cell in the field of view. The red curve illustrates the average response of cells. (**C**) Histogram represents the number of the responding cells with stimulation of 56 mM KCl and 100 µM. The number of responding cells was calculated by determining the number of cells that showed >10% increase in fluorescence intensity compared to that of the baseline. *n* > 50 cells for each solution. (**D**) Representative image illustrated morphology of iSN. Microphotographs of the iMSN obtained with the optimized protocol. Immunofluorescent staining of cells with DAPI, antibodies for TUJ1 (secondary antibodies Alexa 488), DARPP32 (secondary antibodies Alexa 555), and composite image. Confocal microscopy, ×60. The scale bar corresponds to 20 μm.

**Figure 4 ijms-24-06799-f004:**
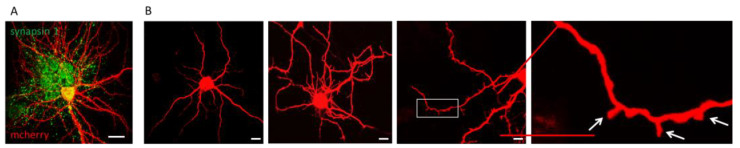
Reprogrammed iSNs form dendritic spines in co-cultures with mouse cortical neurons. (**A**) Representative image of iSN in co-culture with mouse cortical neurons on 14 DIV. iMSNs after the end of the reprogramming procedure were infected with lenti-mcherry. Immunofluorescent staining of co-culture with antibodies to the neuronal marker synapsin 1 (secondary antibodies conjugated with the Alexa 488 fluorophore). Confocal microscopy, ×60. The scale bar corresponds to 10 μm. (**B**) Representative image of reprogrammed iSN infected with lenti-mcherry in the absence of cortical neurons (0 DIV) and in co-culture with mouse cortical neurons at 7 and 10 DIV. White arrows indicate the places of dendritic spine formation. Confocal microscopy, ×60. Scale bar corresponds to 10 μm.

**Table 1 ijms-24-06799-t001:** List of the antibodies used for immunofluorescence.

Title	Vendor	Catalog Number	Dilution
Anti-MAP2 antibody	Abcam	ab281588	1/1000
Anti-β-tubulin III	R&D systems	MAB1195	1/1000
Anti-synapsin I	Chemicon	AB1543P	1/1000
Anti-DARPP-32 antibody	Thermofisher Scientific	MA5-32113	1/250
Alexa Fluor 488 goat anti-rabbit	Invitrogen	R37116	1/1000
Alexa Fluor 555 goat anti-rabbit	Invitrogen	A27039	1/1000

## Data Availability

The raw and analyzed data are available upon request.

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
