# Peer review of "Protocol Optimization for Direct Reprogramming of Primary Human Fibroblast into Induced Striatal Neurons"

_ijms, 2023, doi:10.3390/ijms24076799_

Round 1
Reviewer 1 Report
The objective of the current study was to optimize protocol used for direct reprogramming of primary human fibroblast into induced striatal neurons. According to the study's findings, the authors proposed a protocol to overcome the problem of the presence of multinucleated neurons by an additional synchronize step. Synchronization cell in G1 phase increases the homogeneity of the fibroblasts population, increase survival rate of induced striatal neurons and also eliminate the presence of multinucleated cells at the end of the reprogramming procedure. Moreover, the
induced medium spiny 26 neurons (iMSN) (iMSN) are functionally active and able to form synaptic connections in co-31 culture with mouse cortical neurons.
The research is interesting, had a rational experimental plan, and came to a clear conclusion regarding the proposed new experimental protocol. This paper presented a novel protocol for development of iMSV neurons and provided morphological evidence for the process of development and differentiation. However, provision of some molecular markers (genomics and or proteomics) for the differentiated iMSN particularly in comparison to other neuronal types is essential to prove the molecular identity of the developed iMSN.
Major concerns to address:
Without provision of genomic and proteomic evidence to support the developed iMSC, the validity of the presented morphological data may be questionable. Provision of genomic and proteimic characterization of the developed iMSN at the molecular level is essential and will increase the validity of the results obtained. Using an animal model of GBM, the authors must validate the suggested model at least at the preclinical level. Preclinical models should be created with consideration for including as many GBM-related parameters as were employed in the current investigation such as TMB, TME, ICPIs, OS, etc.
Minor concerns to address:
- Some sentences need to be simple and re-rewetting. For example, Line 45-84.
- Line 140-180: this part need to be re-written as paragraphs not in a numerical order.
- Line 155: PBS (370 C) need to be corrected.
- The quality/resolution of Figure 2 A and B need to be improved.
- Line 429: lentiviral partials?
Author Response
Dear Sir or Madam! Thank you so much for the review. Please find point-by-point response below:
Major concerns to address:
We agree that without genomic and proteomic data the use of the term “induced medium spiny neurons” is not entirely correct. Since it is not possible to conduct such experiments in the short time allotted for responses to reviewers, and considering that the main goal of the work is aimed at reducing multinucleation and increasing the homogeneity of the population, we renamed the middle spiny neurons into induced striatum neurons. We will definitely take into account the remark and make appropriate experiments for the next publication.
Minor concerns to address:
- Some sentences need to be simple and re-rewetting. For example, Line 45-84.
Paragraph 45-84 has been re-written and rearrange
- Line 140-180: this part needs to be re-written as paragraphs not in a numerical order.
We move the detailed protocol in the supplementary section as suggested by another reviewer and correct the numeration in section “Results”
- Line 155: PBS (370 C) need to be corrected.
Corrected
- The quality/resolution of Figure 2 A and B need to be improved.
We replace 2 pictures on Figure 2A and B with poor resolution
- Line 429: lentiviral partials?
Corrected to lentiviral particles

Reviewer 2 Report
In the paper N. Kraskovskaya et al., the authors propose some modifications of a previously developed direct reprogramming protocol, based on the combination of microRNA and transcription factors for obtaining functional induced medium spiny striatal neurons. The modification includes synchronization of the cell cycle in the G1 phase by rapamycin, which leads to a decrease in the number of multinucleated cells by more than 3 times. Kraskovskaya et al. also excluded DLX1 transduction in order to minimize genetic modifications of reprogrammed cells and obtained the same results.
The article is well done and beautifully illustrated and can be accepted with minor corrections.
1. Line 53-55. «Upon transition to the pluripotent state, neuronal cells lose almost all epigenetic marks that were acquired during the differentiation and maturation of somatic cells from which they originate». It seems that iPSC-derived neurons lose all epigenetic marks. In addition to Horvath's article, one can refer to Sheng’s paper in Nature Communicat. regarding epigenetic rejuvenation https://www.nature.com/articles/s41467-018-06398-5
2. A list of abbreviation is needed.
3. Line 103, the first paragraph of the Results. The Result section must not consist a discussion.
4. Line 115-116 repeats line 95 (regarding more than one nucleus).
5. The Protocol in section 2.2 is too detailed (make single use aliquots/thaw aliquots on ice, etc.). The section 2.2. is duplicated in the figure 1E (I would move it into the Suppl).
Author Response
Dear Sir or Madam! Thank you so much for the review. Please find point-by-point response below:
- We add the reference « A stably self-renewing adult blood-derived induced neural stem cell exhibiting patternability and epigenetic rejuvenation" by Sheng’s et al. in line 53-55
- We add list of abbreviation added after Conflict of interest section
- We rearrange the line 115-116 to avoid repetition about multinucleated cells
- We move detailed protocol and information about lentivirus production in Supplementary materials in the end of the manuscript.

Round 2
Reviewer 1 Report
The authot have answered to all my concers